# No Sequestration of Commonly Used Anti-Infectives in the Extracorporeal Membrane Oxygenation (ECMO) Circuit—An Ex Vivo Study

**DOI:** 10.3390/antibiotics13040373

**Published:** 2024-04-19

**Authors:** Hendrik Booke, Benjamin Friedrichson, Lena Draheim, Thilo Caspar von Groote, Otto Frey, Anka Röhr, Kai Zacharowski, Elisabeth Hannah Adam

**Affiliations:** 1Department of Anesthesiology, Intensive Care and Pain Medicine, University Hospital Muenster, University of Muenster, Albert-Schweitzer-Straße 33, 48149 Muenster, Germany; 2Department of Anaesthesiology, Intensive Care Medicine and Pain Therapy, University Hospital Frankfurt, Goethe-University Frankfurt, Theodor-Stern Kai 7, 60590 Frankfurt am Main, Germany; ben@dsgfrankfurt.de (B.F.); lena.draheim@ukffm.de (L.D.); zacharowski@med.uni-frankfurt.de (K.Z.); adam@med.uni-frankfurt.de (E.H.A.); 3Department of Pharmacy, Heidenheim General Hospital, Schloßhaustraße 100, 89522 Heidenheim, Germany; otto.frey@kliniken-heidenheim.de (O.F.); anka.roehr@kliniken-heidenheim.de (A.R.)

**Keywords:** antibiotics, pharmacokinetics, ECMO, critical illness

## Abstract

Patients undergoing extracorporeal membrane oxygenation (ECMO) often require therapy with anti-infective drugs. The pharmacokinetics of these drugs may be altered during ECMO treatment due to pathophysiological changes in the drug metabolism of the critically ill and/or the ECMO therapy itself. This study investigates the latter aspect for commonly used anti-infective drugs in an ex vivo setting. A fully functional ECMO device circulated an albumin–electrolyte solution through the ECMO tubes and oxygenator. The antibiotic agents cefazolin, cefuroxim, cefepime, cefiderocol, linezolid and daptomycin and the antifungal agent anidulafungin were added. Blood samples were taken over a period of four hours and drug concentrations were measured via high-pressure liquid chromatography (HPLC) with UV detection. Subsequently, the study analyzed the time course of anti-infective concentrations. The results showed no significant changes in the concentration of any tested anti-infectives throughout the study period. This ex vivo study demonstrates that the ECMO device itself has no impact on the concentration of commonly used anti-infectives. These findings suggest that ECMO therapy does not contribute to alterations in the concentrations of anti-infective medications in severely ill patients.

## 1. Introduction

A major development in medicine is the aim to achieve more individualized care for patients, including a patient-specific dosage of drugs. While this approach may be straightforward for immediately acting drugs, it may be challenging for drugs exerting a long-lasting effect, such as antibiotics [1]. Individualized dosing often uses therapeutic drug monitoring at different time points, relying on only single parameters influencing drug metabolism, such as kidney function as assessed by the estimated glomerular filtration rate (eGFR), and tables or programs to guide dosing. While this approach might consider kidney function and temporarily altered drug clearance, other factors influencing the pharmacokinetics (PK) of drugs are not considered.

Such factors may manifest in patients requiring intensive care. Most critically ill patients suffer not only from single impaired organ function but combine other factors that might alter PK, such as hypoalbuminemia, multiple organ failure, or capillary leakage (resulting in increased volume of distribution in hydrophilic drugs) [2]. Extracorporeal devices such as extracorporeal membrane oxygenation (ECMO) might further change PK, as the sequestration and binding of drugs to tubes and the oxygenator are known factors potentially influencing PK [1,3,4].

The latter has been demonstrated for several drugs in ex vivo studies [5]. However, for many common antibiotics, it is still unknown whether they bind to surfaces of the ECMO system. Sequestration might put the patient at risk for subtherapeutic antibiotic exposure, and as therapeutic drug monitoring is not ubiquitously available or feasible, this might go unnoticed. Therefore, expanding knowledge on which antibiotics are sequestered in the ECMO circuit is of high interest, and ex vivo studies can help to more precisely address the influence of ECMO on the PK of anti-infective drugs.

This study aims to provide relevant information on the possibility of degradation or adsorption of commonly used anti-infectives in the ECMO circuit.

## 2. Results

The ex vivo circuit ran for 240 min without technical issues or interruptions. A fluid flow of 5 L/min and a temperature of 37 °C were maintained over the whole study period. For each substance, three samples were collected at each of the time points (24 samples per substance). The heated stirrer plate kept the control jar at 37 °C over the entire study period and the control measurements revealed no signs of incompatibility or instability of the substances over time. Table 1 reports the mean concentration and standard deviation of the substances at each given time point. Figure 1 demonstrates relative changes in concentration (continuous graph), with a comparison to the control jar also visualized (dotted graph). The mean drug recovery from the circuit and control jar after 240 min (4 h) was 97% and 107% for cefazolin, 96% and 97% for cefuroxim, 103% and 105% for cefepime, 102% and 100% for cefiderocol, 94% and 97% for daptomycin, 100% and 104% for linezolid and 103% and 102% for anidulafungin. For all drugs combined, a mean recovery of 99.3% in the circuit and 101.7% in the control jar (∆1.4%) was observed after 4 h.

## 3. Discussion

This ex vivo study investigates whether commonly used anti-infective drugs are sequestered in the ECMO circuit. We did not observe a clinically relevant amount of drug sequestration over the study period of 240 min in any of the tested substances.

The correct dosing of anti-infective drugs in vulnerable patients, such as those requiring ECMO therapy, is of great importance, as infections and sepsis are among the most common causes of death in critically ill patients [6]. Since many factors influence the PK of antibiotics and antifungals, identifying an effective dosage for anti-infective agents without causing toxic effects can be challenging. With the development of extracorporeal devices, an additional factor potentially altering the PK has emerged, as drugs can be sequestered in the ECMO circuit [4]. Obtaining data on the potential influence of ECMO on the PK of anti-infectives is of significant clinical relevance, as therapeutic drug monitoring is not routinely available for all drugs and is a labor-intensive process.

Hence, we aimed to provide data for substances for which few (linezolid, daptomycin, anidulafungin) or no data are available (cefiderocol) [7,8]. To our knowledge, this study provides the first ex vivo data on linezolid, anidulafungin and cefiderocol on ECMO addressing potential sequestration. Put into context with inpatient data, subtherapeutic levels of linezolid during ECMO are reported, but our study suggests that other factors (than sequestration) during critical illness contribute to reduced plasma concentrations [9,10,11]. Patient data on anidulafungin during ECMO are rare but deliver no hints of altered PK [8]. A comparison to PK of cefiderocol in ECMO patients is not possible as no data are available.

Altogether, the influence of pharmacologic properties on sequestration in the ECMO circuit is unknown or controversial [12,13,14]. However, a potential impact of lipophilicity and protein binding is suggested. We chose drugs with different pharmacologic properties or with similar properties but differences in one aspect, e.g., protein binding (cefazolin, cefuroxim and cefepime). In our study, protein binding had no impact on sequestration, as drugs with high variability in protein binding (20% for cefepime, 50% for cefuroxim and 74–86% for cefazolin) showed no difference in sequestration in our experimental setup. This suggests that factors other than protein binding may contribute to sequestration. Furthermore, other properties, such as molecular weight and lipophilicity, had no impact in our study, as linezolid (small) compared to daptomycin (large) and anidulafungin (lipophilic) compared to cephalosporins (hydrophilic) all showed no relevant reduction in concentration during the study period.

The difference in the ex vivo results may be explained by the varying setups of studies. First, different fluids (e.g., whole blood vs. crystalloids vs. albumin solutions) have been shown to impact results. For example, a more pronounced loss for ampicillin was found in crystalloid-primed circuits than in blood-primed circuits [15]. Second, different devices (oxygenators and pumps) can cause variable results for the same drugs. As a result, different extents of sequestration were found for centrifugal pump circuits with hollow-fiber membrane oxygenators compared to neonatal roller pump circuits with silicone membranes [14,16,17]. Third, it remains to be identified whether drugs may directly bind to the circuit or whether drugs bind to circuit-bound blood proteins. Differences between used and new circuits have been observed and suggest that ECMO circuits show a level of saturation with a greater reduction in concentration after the change in oxygenators [18]. Fourth, a huge percental loss in a small test fluid does not necessarily result in clinical relevance in cases of observed sequestration. Our approach uses a test fluid volume that is close to the volume of distribution of the tested drugs. In doing so, the absolute amount of substance added is equal to typical clinical practice and percental losses are easier to interpret.

Predicting pharmacokinetics in patients with ECMO remains difficult, as no clear factor contributing to sequestration can be identified. Although protein binding and lipophilicity were suggested, other studies, including ours, were not able to confirm this impact on sequestration. However, in vivo coherences are more complex. Changes in blood pH and the general environment (of albumin) might have an impact on protein binding and could change the sequestration behavior of these drugs [19]. In addition, protein–protein interactions over longer periods of ECMO treatment might become more relevant as certain proteins start to attach to the circuit membrane or tubes [20]. Nonetheless, our study suggests that sequestration in the ECMO circuit is negligible, especially in comparison to other factors affecting PK. The observation of impaired renal function, abnormal protein levels, or volume shifts is common in critically ill patients, and these are known factors to have an impact on PK [2]. For the accurate identification of factors influencing drug pharmacokinetics and to enable dose adjustments tailored to individual patient characteristics and ECMO treatment, comprehensive inpatient data for the drugs under investigation are essential.

This study has potential limitations. First, we opted for an albumin–electrolyte test fluid to allow for a larger volume of test fluid. This limits the results’ comparability to other studies that mostly used whole blood test fluids but smaller volumes of test fluid. Second, our study had an observation period of 4 h, which differs from other studies employing a 24 h observation. However, whenever sequestration occurred in an ex vivo setup, it was already observable in the first hours of the experiment [21]. Nonetheless, we might have missed sequestration at a later time point.

## 4. Materials and Methods

This is an experimental ex vivo study aiming to evaluate whether commonly used anti-infective drugs are sequestered in the ECMO circuit. This study was performed at the laboratory units of the university hospital in Frankfurt, Germany and measurements took place in the laboratory units of the Department of Pharmacy at the Heidenheim General Hospital in Germany. Ethics committee approval was not required due to the solely experimental setup which did not require patient or animal material. The study was funded as part of a project supported by the German Research Foundation (DFG; AD 592/1-1).

### 4.1. Drug Selection

This experiment aimed to test antibiotics and antifungals commonly administered in patients receiving ECMO therapy. Furthermore, we aimed to examine substances with different proportions of protein binding, molecular weights and lipophilicity to address whether these have an impact on adsorption. Therefore, we included drugs that mainly differ in protein binding such as cefazolin (protein binding 74–86%), cefuroxim (protein binding 50%) and cefepime (protein binding 20%). Cefiderocol was included in our analysis as it is a rather novel substance, and little is known about it in the context of ECMO treatment. Linezolid, an oxazolidinone, and daptomycin, a cyclic lipopeptide, are the smallest and the largest drug molecules here (337.35 g/mol vs. 1619.71 g/mol). The antifungal drug anidulafungin was also added and is mostly lipophilic (logP 2.9). All physicochemical properties were obtained from DrugBank^®^ and/or the corresponding product information [22]. For details, see Table 1.

### 4.2. Experimental Setup

The experimental setup contained a fully functional ECMO device (Cardiohelp System, Getinge AB, Gothenburg, Sweden), including a polymethylpentene oxygenator and pump (HLS Set Advanced, Getinge AB, Gothenburg, Sweden) standard length of heparin coated tubes (Getinge AB, Gothenburg, Sweden), a HU-35 heater unit (Getinge AB, Gothenburg, Sweden) (keeping fluids at 37 °C), a 25 L glass container, human albumin 20% (20 g/100 mL), crystalloids, and the aforementioned six antibiotics, as well as one antifungal drug (see Figure 1). A smaller jar (total volume of 100 mL) with the same solution and the same drugs but without ECMO was used to demonstrate drug compatibility and stability over the planned time of the experiment. This jar was kept at 37 °C on a heated stirrer plate.

### 4.3. Experiment

A circulating volume of 20 L was determined to mimic a realistic volume of distribution (Vd) close to the Vd of the tested drugs in a patient (see Table 1). In order to detect the potential influence of plasma protein binding, 4 L of 20% human albumin and 16 L of crystalloids were mixed to achieve an albumin–electrolyte-solution with a physiological albumin concentration of 4 mg/dL.

The amount of added anti-infectives was calculated to achieve concentrations that are comparable to in vitro plasma concentrations (see Table 2).

Three baseline samples were acquired before ECMO, primed with 500 mL of crystalloids, was initiated. ECMO fluid flow was set to 5 L/min and the heater unit was set to maintain a temperature of 37 °C. This experiment was conducted for 4 h and three samples of 1 mL each were taken at 5, 15, 45, 60, 120, 180 and 240 min. The samples were frozen immediately at −80 °C and shipped to the laboratory of the pharmacy department of Heidenheim General Hospital, Germany, for analysis.

### 4.4. Bioanalytical Methodology

Three samples were analyzed at each time point using validated high-performance liquid chromatography (HPLC) assays with ultraviolet detection (Nexera-I 3D plus, Shimadzu, Kyoto, Japan). This was performed in line with the Valistat 2.0 (ARVECON GmbH, Walldorf, Germany) validation criteria as required by the German Society of Toxicology and Forensic Chemistry (GTFCh), which are routinely used in the context of therapeutic drug monitoring [23,24]. Calibration curves were linear over the concentration range. Relative standard deviations for accuracy and precision were all <10% and within the acceptable limits of ±15% required by the validation criteria. The same setup is used to analyze clinical samples from patients and is used in daily practice [24,25].

All drugs were obtained as a regular vial containing powder or solutions (linezolid) and the powders were dissolved according to the pharmaceutical information. For calibration standards (CS) and quality controls (QS), the 4% albumin solution was spiked directly with aliquots of the stock solution and immediately frozen at −80 °C. Albumin precipitation was performed by adding 200 µL of an acetonitrile/methanol (1:1) mixture. This mixture contained metronidazole (for cephalosporins, daptomycin and linezolid) or midazolam (for anidulafungin) as an internal standard. Exemplary results of the HPLC assays are presented in Figure 2, Figure 3 and Figure 4.

### 4.5. Concentration Analysis

For analysis, we used the mean value of the three samples of each time point. Concentrations were set in reference to the baseline concentration and are reported as a percentage of that baseline concentration. The mean value of the three samples was calculated for analysis. We assessed whether a reduction in concentration occurred for the investigated drugs over the course of the experiment.

## 5. Conclusions

Our study provides new insights into pharmacokinetics (PK) during ECMO, presenting ex vivo data for the first time on cefiderocol, linezolid, and anidulafungin. We found no evidence of drug properties contributing to sequestration, such as protein binding or lipophilicity. Our results demonstrate that cefazolin, cefuroxim, cefepime, cefiderocol, linezolid, daptomycin, and anidulafungin are not sequestered in an ex vivo setting. These findings suggest that dose adjustments based on ECMO therapy alone may not be necessary. However, other effects of ECMO treatment on PK still need careful consideration.

## Figures and Tables

**Figure 1 antibiotics-13-00373-f001:**
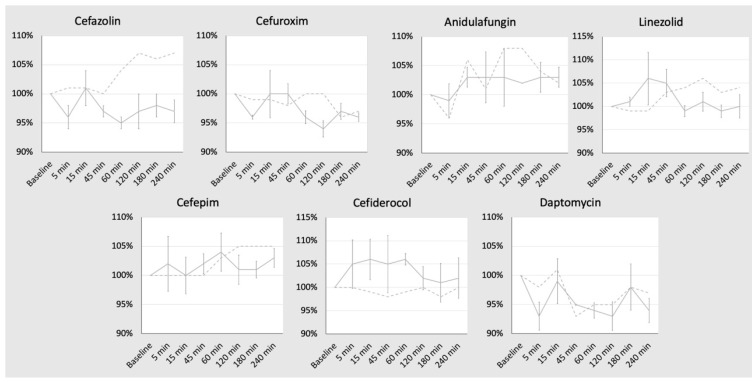
Each diagram shows the change in concentration over time for the substance tested. The *x*-axis shows the time at which the probes were taken. The *y*-axis shows the mean percentage (±SD) of drug recovery relative to the baseline concentration at each time point. The baseline concentration is the starting point of each graph. The continuous graph represents the change in concentration of the ECMO circuit, while the dotted graph is the change in concentration of the control. min: minutes.

**Figure 2 antibiotics-13-00373-f002:**
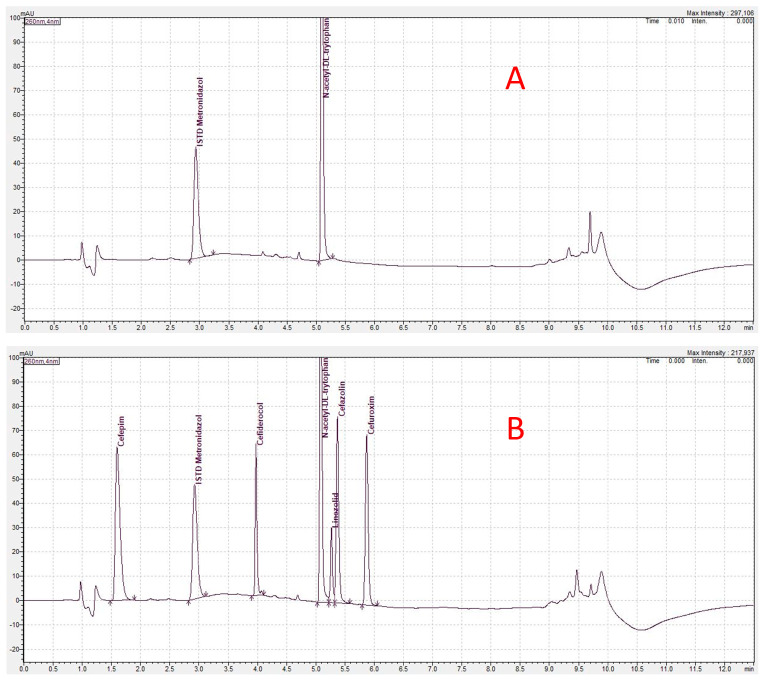
High-performance liquid chromatography result for cephalosporins and linezolid. The *x*-axis shows the retention time in minutes and the *y*-axis shows the adsorption of detection light at 260 nm wavelength. (**A**) Result of the probe with the internal standard without antibiotics added. Two spikes are displayed. The left spike is the internal standard metronidazole and the right spike is N-acetyl-DL-tryptophan, which is used as a stabilizer added to human albumin. (**B**) Result from an exemplary sample of the ex vivo study. The internal standard, N-acetyl-DL-tryptophan and cephalosporin and linezolid adsorption at 260 nm wavelength are displayed.

**Figure 3 antibiotics-13-00373-f003:**
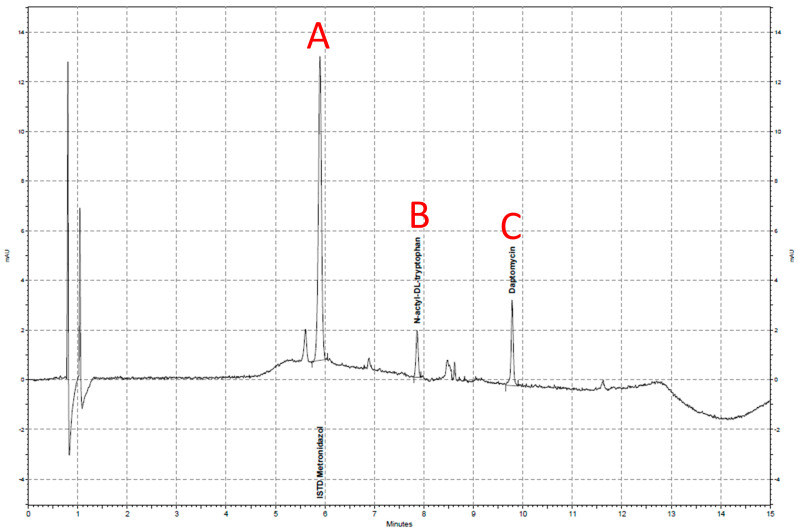
High-performance liquid chromatography result for daptomycin. The *x*-axis shows the retention time in minutes and the *y*-axis shows the adsorption of light at a 350 nm wavelength. Three spikes are shown: metronidazol (A; internal standard), N-acetyl-DL-tryptophan (B; human albumin stabilizer) and daptomycin (C).

**Figure 4 antibiotics-13-00373-f004:**
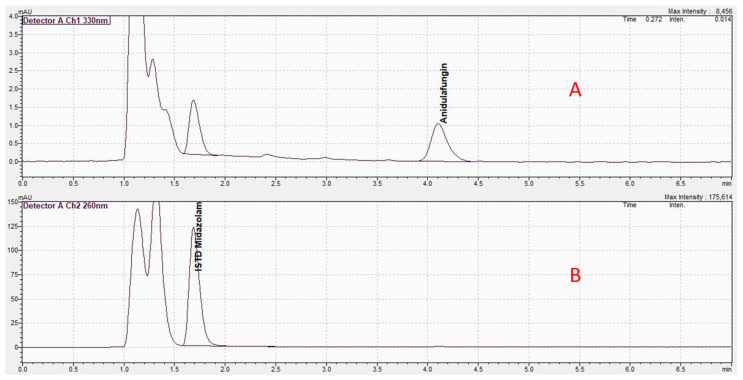
High-performance liquid chromatography result for anidulafungin. The *x*-axis shows the retention time in minutes and the *y*-axis shows the adsorption of light at a 330 nm (**A**) and 260 nm (**B**) wavelength. (**A**) Result of an exemplary probe of the ex vivo study. The spike to the right shows the adsorption of light for anidulafungin at 330 nm, whereas the left spike at 260 nm represents the internal standard (midazolam). (**B**) Result of a probe without anidulafungin. Only the spike for the internal standard midazolam is shown.

**Table 1 antibiotics-13-00373-t001:** Target concentrations, baseline concentrations and concentrations over time of the tested substances. The mean concentrations of the three probes per time point and their standard deviation are given. Abbreviations: L: liter; mg: milligram; min: minute.

Substance	Target Concentration (mg/L)	Baseline (0 min) (mg/L)	5 min (mg/L)	15 min (mg/L)	45 min (mg/L)	60 min (mg/L)	120 min (mg/L)	180 min (mg/L)	240 min (mg/L)
Cefazolin	100	89.6 ± 1.2	85.6 ± 2.3	90.6 ± 2.6	86.9 ± 1.4	84.9 ± 1.6	86.5 ± 2.2	88.1 ± 1.8	86.6 ± 1
Cefuroxim	75	72.3 ± 1	69.5 ± 0.8	72.5 ± 2	72.2 ± 2.1	69.3 ± 1.3	67.8 ± 0.5	69.8 ± 1.6	69.4 ± 0.5
Cefepime	100	100.3 ± 1.7	102.3 ± 3	100.5 ± 1.5	102.4 ± 1.6	104.4 ± 1.6	101 ± 2.4	101.1 ± 2.9	103.7 ± 1.3
Cefiderocol	50	51.9 ± 1.4	54.8 ± 2	55.2 ± 1.6	54.4 ± 2.8	55 ± 1.7	53 ± 1.5	52.6 ± 1.2	53.1 ± 0.7
Linezolid	15	14.3 ± 0.2	14.5 ± 0.2	15.2 ± 0.7	15 ± 0.4	14.2 ± 0.1	14.5 ± 0.3	14.1 ± 0.3	14.3 ± 0.2
Daptomycin	25	25.3 ± 0.5	23.6 ± 0.5	25.2 ± 0.8	24.1 ± 0.5	23.8 ± 0.2	23.6 ± 0.6	24.8 ± 0.5	23.8 ± 0.1
Anidulafungin	5	4.80 ± 0.1	4.75 ± 0.1	4.95 ± 0.1	4.95 ± 0.1	4.95 ± 0.2	4.90 ± 0.1	4.95 ± 0.2	4.95 ± 0.1

**Table 2 antibiotics-13-00373-t002:** Characteristics of tested substances. Abbreviations: L: liter; mg: milligram; kg: kilogram; g: gram; h: hour; Vd: volume of distribution.

Substance	Molecular Weight [g/mol]	logP	Protein Binding [%]	Vd [L/kg]	Vd Patient 70 kg Bodyweight [L]	Usual Bolus Dosage [mg]	Calculated c_max_ [mg/L]	Calculated c_4h_ [mg/L]	Aimed Concentration [mg/L]	Amount Added [mg]	T_1/2_ Normal Patient [h]	Clearance Normal Patient [L/h]	Qo (Non Renal Clearance)
Cefazolin	454.51	−0.58	74–86	0.17	11.9	2000	168.1	42.0	100	2000	2	4.1	0.1
Cefuroxim	424.39	−0.16	50	0.2	14	1500	107.1	18.9	75	1500	1.6	6.1	0.1
Cefepime	480.56	−0.37	20	0.27	18.9	2000	105.8	26.5	100	2000	2	6.5	0.15
Cefiderocol	752.21	−2.27	40–60	0.26	18	2000	111.1	36.8	50	1000	2.5	5.2	0.03
Linezolid	337.35	0.9	31	0.65	45.5	600	13.1	7.7	15	300	5.1	6.2	0.7
Daptomycin	1619.71	−0.47	90–94	0.1	7	350	50	36.1	25	500	8.5	0.2	0.5
Anidulafungin	1140.2	2.9	84	0.5	35	200	5.7	5.1	5	100	24	1.0	0.99

## Data Availability

The raw data supporting the conclusions of this article will be made available by the authors upon request.

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
