# Peer review of "No Sequestration of Commonly Used Anti-Infectives in the Extracorporeal Membrane Oxygenation (ECMO) Circuit—An Ex Vivo Study"

_antibiotics, 2024, doi:10.3390/antibiotics13040373_

Round 1

Reviewer 1 Report

Comments and Suggestions for Authors

I read this manuscript with great interest given the scarcity of data regarding pharmacokinetics in ECMO. The authors investigated 6 antibiotics and 1 antifungal substance in an ex-vivo model of an ECMO circuit running for 6h. For the first time they showed nearly unchanged concentrations over a 6h period of these drugs. Morevoer, the choice of the drugs seems to be a well educated guess.

This well written manuscript is of great interest for intensivists prescribing anti-infective drugs for patients during ECMO support. However, several points need to be clarified:

Title: the title should be more precise, you could state the ex-vivo approach. Moreover, you investigated 6 antibiotics and 1 antifungal substance.

Discussion: could you give the readership your point of view what effects may alter pharmacokinetics in ECMO as protein binding, lipophilicity and molecular weight did not.

Furthermore, I wondered if the used circuit tubes were heparin-coated?  Please clarify in the methods.
Moreover, you stated the different ex-vivo findings when using hollow-fibre membrane oxygenators or silicone membranes. Please add to the methods section the kind of oxygenator employed.

Minor comment:
The readability of the legends of the respective panels of figure 1 is poor. I believe this is a technical issue due to the pdf, isn´t it?
The panels with the 7 drugs could be greater or rather with a greater solution.

Comments on the Quality of English Language

The English is fine.

Reviewer 2 Report

Comments and Suggestions for Authors

In the current study entitled “No sequestration of commonly used antibiotics is observed in extracorporeal membrane oxygenation” the authors studied the effect of ECMO device medication levels in critically ill patients. They set up a functioning ECMO circuit outside the body (ex-vivo) and circulated an albumin-electrolyte solution, mimicking blood, through the tubes and oxygenator. They then added common prophylaxes (cefazolin, cefepime, cefiderocol, linezolid, daptomycin) and an antifungal medication (anidulafungin) to the solution. Over four hours, they regularly extracted blood samples and measured drug concentrations using HPLC UV detection. Their analysis revealed no significant changes in the concentration of any medications throughout the experiment. This study offers a valuable initial exploration into ECMO's impact on pharmacokinetics of prophylaxis and antifungals. However, to strengthen the study for publication in Antibiotics, additional methods for validating the HPLC data are recommended. Here are two suggestions.

1.       Use another Quantitative method such as Nuclear Magnetic Resonance (qNMR) that could provide complementary data on drug concentrations, enhancing the reliability of the HPLC findings. OR

2.      Doing some qualitative analysis such as Measuring the Minimum Inhibitory Concentration (MIC) of the antibiotics at different time points would provide additional evidence regarding the reproducibility of HPLC data.

Minor concerns:

1. please improve the quality and resolution of Figure1

2 Please add a little more details in the results.

Reviewer 3 Report

Comments and Suggestions for Authors

The authors provide a paper on an important and timely topic. However, there are several aspects requiring careful consideration.

Investigations have demonstrated that ECMO leads to significant changes in PK. The accumulation of blood proteins/cells on ECMO and changes concerning the plasma protein components have been proposed as a contributing factor to these problems. The paper lacked describing possible changes (during in-vivo therapy!) regarding: 1. Plasma proteins composition and 2. Mechanisms of protein bindings (alterations in binding can cause major changes in available free or active drug).

Extend the Chromatographic analysis section: sample preparation, add HPLC-protocol and representative HPLC-chromatogram.

Moreover please reconsider your title and conclusion: “No sequestration of commonly used antibiotics…” and “These findings suggest that dose adjustments based solely on ECMO therapy may not be necessary for these drugs.”

Without comparing in-vivo data with TDM and a longer observation period, this statements are not justified and may lull physicians into a false sense of security!

Round 2

Reviewer 1 Report

Comments and Suggestions for Authors

Thank for your expeditious responses to my questions. I feel the manuscript now suitable for publication.

Reviewer 2 Report

Comments and Suggestions for Authors

I was unable to check the material method and intro section please check it carefully and remove the mistakes/typos (if any)

Reviewer 3 Report

Comments and Suggestions for Authors

Just one question: Why retention time of internal standards e.g. metronidazole (Fig.2. B versus Fig.3. A) changed?

Flow modified?